# The Influence of Gingival Exposure on Smile Attractiveness as Perceived by Dentists and Laypersons

**DOI:** 10.3390/medicina58091265

**Published:** 2022-09-13

**Authors:** Bianca Maria Negruțiu, Andrada Florina Moldovan, Claudia Elena Staniș, Claudia Teodora Judea Pusta, Abel Emanuel Moca, Luminița Ligia Vaida, Cristian Romanec, Ionut Luchian, Irina Nicoleta Zetu, Bianca Ioana Todor

**Affiliations:** 1Department of Dentistry, Faculty of Medicine and Pharmacy, University of Oradea, 1 University Street, 410087 Oradea, Romania; biancastanis@yahoo.com (B.M.N.); ligia_vaida@yahoo.com (L.L.V.); biancaioana.todor@gmail.com (B.I.T.); 2Department of Clinical Disciplines, Faculty of Medicine and Pharmacy, University of Oradea, 1 University Street, 410087 Oradea, Romania; onita.andrada@yahoo.com (A.F.M.); stanis.claudia@yahoo.com (C.E.S.); 3Department of Morphological Disciplines, Faculty of Medicine and Pharmacy, University of Oradea, 1 University Street, 410087 Oradea, Romania; 4Department of Orthodontics and Dentofacial Orthopedics, Faculty of Dentistry, “Grigore T. Popa” University of Medicine and Pharmacy, 16 Universității Street, 700115 Iași, Romania; cromanec@gmail.com; 5Department of Periodontology, Faculty of Dentistry, “Grigore T. Popa” University of Medicine and Pharmacy, 16 Universității Street, 700115 Iași, Romania; ionut_luchian@yahoo.com

**Keywords:** gingival exposure, Romanian dentists, Romanian laypersons, smile attractiveness

## Abstract

*Background and Objectives*: Smile is an important mark of beauty, and smile attractiveness can be influenced by various factors, one of these being the amount of gingival exposure. The aim of this research was to evaluate the perception of an ideal gingival exposure in smile in a sample of Romanian dentists and laypersons, and to find out what is the most important aspect that influences the smile perception for the respondents included in the study sample. *Materials and Methods*: An online survey was conducted between 3 February 2020 and 31 October 2020. The authors developed a short questionnaire consisting of 7 items. The first four items investigated the respondents’ age, gender, profession and whether or not they underwent an orthodontic treatment in the past. For items 5 and 6, participants had to choose the most and the least attractive amount of gingival exposure, and for the last item they had to choose the factor that mostly influences the smile attractiveness in their opinion. *Results*: 235 questionnaires remained in the study. The sample consisted of 194 women and 41 men, 162 laypersons and 73 dentists. The average value for the most attractive amount of gingival exposure was −0.57 ± 2.407 mm, with a median value of 0 mm while the average value for the least attractive amount of gingival exposure was 1.43 ± 3.785 mm, with a median value of 4 mm. The differences between the most or least attractive gingival exposure perceived by the participants were not significant according to gender or professional category (*p* > 0.05), but, when compared between having or not having an orthodontic treatment in the past the differences were statistically significant (*p* < 0.05). As such, participants who had an orthodontic treatment in the past perceived a significantly higher value for the most attractive gingival exposure, and a significantly higher value for the least attractive gingival exposure (*p* = 0.026) than the participants who didn’t have an orthodontic treatment in the past. As for the factor that mainly influences smile attractiveness, laypersons chose significantly more frequent white teeth or aligned teeth (96%, 71.7%) while dentists chose significantly more frequent a gingival exposure between 0–3 mm (70.4%) as smile factors for an aesthetic smile (*p* < 0.001). *Conclusions*: In this study sample, the respondents considered that the most attractive smile involved a covering of 0.57 mm (in average) of the upper incisors by the upper lip. Although smile attractiveness did not appear to be influenced by gender or profession in this study population, it was influenced by previous orthodontic treatment. The participants’ roles of dentist or layperson influenced the factors chosen for an aesthetic smile.

## 1. Introduction

A person’s attractiveness is mainly conditioned by facial appearance [1], and facial attractiveness depends on several characteristics. Facial symmetry [2], as well as the averageness of the facial features are generally considered factors that contribute to increasing the good aesthetic appearance of a face [1]. When it comes to interpersonal relationships, people are generally attracted either by the eyes or by the mouth of the interlocutor, these being the main facial elements that draw attention [3].

Smile is an important mark of beauty, so much so that people who have a beautiful and harmonious smile are considered more attractive, smarter and even more popular [4]. The aesthetics of the smile is considered paramount, most people believing that a beautiful smile is the second most important element of facial beauty, after the eyes [5].

Although, smile evaluations are usually subjective, the aesthetics of the smile can be evaluated objectively, as well, using the diagram of facial aesthetic references (DFAR) which consists of six lines drawn around maxillary incisors and canines. The DFAR shows the position of the teeth and the ratio between the teeth, as well as the relationship between the lip and gingiva [6,7].

An altered relation between the upper lip and gum can determine excessive gingival exposure. Excessive gingival exposure, also known as gummy smile, can be caused by a large number of factors or even a combination of them, such as: upper lip morphology, altered passive eruption, excessive vertical maxillary growth by excessive growth of the lower third of the face or lip incompetence, heredity, congenital and acquired factors as a result of drug consumption, systemic causes (e.g., hormonal imbalances), general diseases (e.g., leukemia), orthodontic appliances or bacterial plaque [8,9,10,11].

Smile attractiveness can be positively or negatively influenced by the gingival display [12], an excessive gingival exposure usually requiring an appropriate therapeutic approach [13]. Lip-repositioning surgery is considered a safe and predictable approach for the treatment of excessive gingival exposure [13], as is the injection of botulinum toxin A [14]. Orthodontic treatment is another viable option for correcting excessive gingival exposure, improving smile aesthetics and obtaining an ideal occlusion at the end of the treatment [15].

However, the patient’s perception of self should be of utmost importance when it comes to treatment decisions and aesthetic judgment [16], mainly because dentists and laypersons may have different perceptions on smile attractiveness [17]. The premise of this study was the fact that the Romanian population may have a different perception over the degree of gingival exposure considered aesthetically acceptable compared to other populations, and that this perception may vary in regards to patients’ age, gender, and between dentists and laypersons.

This study is of great importance in clinical practice for dentists, especially Romanian dentists, because patients and dentists tend to have different opinions on how a smile should look alike compared to dentists. The latter must take into account and put into practice patients’ wishes so as the patients are pleased with their smile at the end of the treatment. Moreover, this is the first research that highlights the effect of an increased gingival exposure on smile attractiveness from Romanian lay-persons’ and dentists’ point of view. The aim of this research was to evaluate the perception of an ideal gingival exposure in smile in a sample of Romanian individuals, and to compare the perception of an ideal gingival exposure between the various categories of people investigated. We also wanted to find out what is the most important aspect that influences the smile perception for the respondents included in the study sample.

## 2. Materials and Methods

### 2.1. Ethical Considerations

The study was conducted in accordance with the World Medical Association (WMA), Declaration of Helsinki—Ethical Principles for Medical Research Involving Human Subjects, and approved by the Ethics Committee of the University of Oradea, Romania (No. 8/20.01.2020).

### 2.2. Participants and Data Collection

The online cross-sectional survey was conducted between 3 February 2020 and 31 October 2020, and the questionnaires were distributed using the online platform iSondaje.ro. The link provided by the platform was copied, and distributed via social platforms or e-mail.

For this study the authors developed a short questionnaire consisting of 7 items. The language used was Romanian. At the beginning of the questionnaire the respondents were informed that by continuing to complete the survey they confirmed their willingness to anonymously participate in this research, and that they can withdraw from completing the questionnaire at any given time. No time limit was imposed. The respondents’ names were not registered.

The first four items investigated the respondents’ age (the respondents had to type in their age), gender (the respondents had to select from the two available options, male or female), profession (the respondents had to select from the two available options, these being dentist or layperson), and whether or not the respondents’ underwent an orthodontic treatment (the respondents had to choose from the two available options, yes or no).

Items 5 and 6 investigated participants’ perception of smile attractiveness with different degrees of gingival exposure. For a visual illustration both items displayed the same picture that represented a smile with ten values of gingival exposure (from +4 mm to −5 mm). In item 5 the respondents were asked to choose the most attractive smile from the ten displayed options, while in item 6 they were asked to choose the least attractive smile from the displayed options. The intraoral photography was obtained from a 30-year old female patient with ideal occlusal relationships and the head in a natural position. The initial photo was performed with a maximum gingival exposure, the other pictures being edited through incisal movement of the upper lip using Adobe Photoshop 2020 (Adobe Inc., San Jose, CA, USA). The patient gave her written consent to be photographed, and have her photo used for the completion of this research (Figure 1).

The last item consisted of the following question “Which of the following mostly influences the smile, in your opinion?”, and had seven possible answers, these being aligned teeth, white teeth, symmetrical smile, diastema, gingival exposure of 0–3 mm, gingival exposure over 3 mm, and gingival exposure under 3 mm.

Since this was an online survey, only questionnaires that were entirely completed were registered by the online platform. For each item, only one option could be selected. We excluded only questionnaires that were completed by respondents younger than 18 years old.

### 2.3. Statistical Analysis

The results obtained were included in tables using Microsoft Excel 2013 (Microsoft, Redmond, Washington, DC, USA). The data analysis was made using IBM SPSS Statistics 25 (IBM, Chicago, IL, USA). Quantitative variables were tested for normal distribution using the Shapiro-Wilk Test and were written as averages with standard deviations or medians with interquartile ranges. Categorical variables were written as counts or percentages. Quantitative variables with non-parametric distribution were tested using Mann-Whitney U tests. Categorical variables were tested using Fisher’s exact tests. A value of *p* < 0.05 was considered statistically significant.

## 3. Results

The questionnaires were initially completed by a total of 274 respondents, but after applying the exclusion criteria (the respondents had to be older than 18 years old, of Romanian ethnicity, access to social media, with or without dental studies(if they are dentists, minimum 5 years experience), who might need orthodontic treatment and seem interested in improving their facial appearance), 39 questionnaires were eliminated, and only a total of 235 questionnaires were left in this research.

### 3.1. Socio-Demographic Characteristics of the Study Sample

Data from Table 1 shows the characteristics of the studied group. Most of the participants were women (82.6%), and most of the participants were laypersons (68.9%). Only 41.3% of the participants had an orthodontic treatment in the past.

### 3.2. Attitude towards the Ideal Amount of Gingival Exposure

The average value for the most attractive amount of gingival exposure was −0.57 ± 2.407 mm, with a median value of 0 mm while the average value for the least attractive amount of gingival exposure was 1.43 ± 3.785 mm, with a median value of 4 mm (Table 2). Data from Table 3 shows the comparison of the most/least attractive amount of gingival exposure between groups. Both of the parameters were described as having a non-parametric distribution according to the Shapiro-Wilk test in all the studied groups (*p* < 0.05). As such, it can be seen that the differences between the most or least attractive gingival exposure perceived by the participants were not significant according to gender or professional category (*p* > 0.05), according to the Mann-Whitney U tests. But, when compared between having or not having an orthodontic treatment in the past the differences were statistically significant (*p* < 0.05). As such, participants who had an orthodontic treatment in the past perceived a significantly higher value for the most attractive gingival exposure (median = 1 (IQR = −2–2) vs. median = 0 (IQR = −3–1) (*p* = 0.009)) and a significantly higher value for the least attractive gingival exposure (median = 4 (IQR = 0.75–4) vs. median = 4 (IQR = −5–4) (*p* = 0.026)) than the participants who didn’t have an orthodontic treatment in the past.

### 3.3. Attitude towards the Factors That Mostly Influence Smile Attractiveness

Of all the factors that could possibly influence a smile (Item 7), most of the participants agreed that aligned teeth (45.1%), white teeth (21.3%) or a symmetrical smile (17.4%) are the most important factors for an esthetic smile (Table 4).

Data from Table 5 shows the distribution of the participants between smile factors and other parameters. According to Fisher’s tests, it can be seen that the differences of frequencies between chosen smile factors for an aesthetic smile weren’t significant according to gender or existence of an orthodontic treatment in the past (*p* > 0.05). However, data from the comparison of laypersons and dentists and the Z-tests with Bonferroni correction show that they chose significantly differently, as such:Laypersons chose significantly more frequent white teeth or aligned teeth (96%, 71.7%), while dentists chose significantly more frequent a gingival exposure between 0–3 mm (70.4%) as smile factors for an aesthetic smile (*p* < 0.001);The same applies for female participants: laypersons chose significantly more frequent white teeth or aligned teeth (94.9%, 74.2%), while dentists chose significantly more frequent a gingival exposure between 0–3 mm (70.8%) as smile factors for an aesthetic smile (*p* < 0.001);As for male participants: laypersons chose significantly more frequent white teeth (100%), while dentists chose significantly more frequent a gingival exposure between 0–3 mm or a gingival exposure below 0 mm (66.7%, 100%) as smile factors for an aesthetic smile (*p* = 0.017).

## 4. Discussion

In recent years, the aesthetic requirements of the patients who address the orthodontist have considerably risen, thus determining physicians to offer particular importance not only to the alignment of the teeth, but also to the gingival morphology concerning the degree of gum exposure, gingival contour, zenith position and the presence of gingival papillae. Even though the adult patients who address an orthodontist are very compliant compared to children, they are always a challenge. Thus, a complex, interdisciplinary treatment and the use of various treatment methods of patients with dento-maxillary anomalies is justified in order to improve the index of life quality. The increase of the index of life quality implies an improvement of the self-esteem, social self-esteem, performances and the global self-related current thoughts [5,18,19,20,21].

The smile aesthetic expectations of the patients may vary individually. We have noticed that more and more patients are unhappy with the aesthetic of their smile regarding excessive gingival exposure, and are concerned whether or not their gummy smile can be corrected. In order to be able to set a proper diagnosis of gummy smile, the dentist must evaluate the gingival level. Liebart et al. (2004) proposed a classification of the gummy smile considering the gums [22,23,24].

Several criteria about the anterior gingival exposure considered aesthetically acceptable have been established. Morley et al. (2001) considered that only a gingival exposure of 1–3 mm can be considered aesthetic [7], while Kokich et al. (2006) stated that a gingival exposure of up to 3 mm is aesthetically acceptable [25]. On the other hand, Geron et Atalia (2005) have shown that only a maximum of 1 mm gingival exposure can be considered aesthetic [26]. As far as the posterior gingival smile is concerned, no clear rules regarding the degree of gingival exposure have been settled. Some studies have concluded that posterior gingival smile influences more negatively the perception of professionals and laypersons over a smile. Also, professionals and laypeople consider that a posterior gingival exposure of up to 6 mm can be considered aesthetic [4,27].

In our study, the attractiveness of a smile with various degrees of gingival exposure from a maximum of 4 mm gingival exposure (+4 mm) to a incisor coverage by the upper lip of 5 mm (−5 mm) was compared. These limits were set after a proper literature review, in previous studies, gingival exposure varying from −2 mm to +4 mm [28] or from −4.6 mm to +3.3 mm [26] except for Ioi et al. (2010) where gingival exposure varied between +5 mm and −5 mm [12]. Thus, this research is within the limits imposed by previous studies.

To the present day, there is insufficient data concerning the influence of gingival smile on smile attractiveness to the Romanian population. This is the first research that highlights the effect of an increased gingival exposure on smile attractiveness from laypersons’ and dentists’ point of view.

Overall, 0 mm gingival exposure was considered to be part of the most attractive smile, while a gingival exposure of 4 mm was considered to describe the least attractive smile. These results are similar to the conclusions highlighted by Ioi et al. (2010) [12]. Both groups, laypersons and dentists, seemed to agree that the most attractive smile shows 0 mm gingival exposure, while the least attractive smile can be described by a gingival exposure within a range of −4 and +4 mm. In the research of Thomas et al. (2011) it is stated that orthodontists are more critical when assessing a smile attractiveness than dentists and lay persons, at the same time emphasizing that all three study groups considered that reduced gingival exposure is less attractive [29]. Compared to the article published by Geron and Atalia (2005) which points out that lay persons believe that an overlay of the upper incisors by the upper lip of 0–2 mm is the most aesthetic [26], in our study laypersons consider that a 0 mm gingival exposure is the most aesthetic. These results may be influenced by the fact that greater gingival exposure is correlated with youth and joviality.

Pithon et al. (2013) concluded that a maximum gingival exposure does not always affect the aesthetic aspect of a smile, also highlighting that gingival exposure can positively influence the perception of a smile, but only when exposed less. However, an insufficient exposure of maxillary incisors may be considered less attractive [30], opposed to the results obtained and published by Ioi et al. (2010) who underlines that laypersons consider an overlay of the upper incisors by the upper lip of 0–2 mm to be the most aesthetic [12]. These results are similar to the results of this study, Romanian laypersons considering a range between −3 and +1 mm of gingival exposure as being the most attractive.

Cracel-Nogueira et al. (2013) emphasized that a medium smile with minimal gingival exposure was the most attractive and the smile with a large gingival exposure and diastema was considered the least aesthetic. The gender of the respondent did not influence their results, except for the gingival smile when young respondents granted increased values. Taking into consideration the academic training, similar to the results of our study, the responses were very similar within both groups, but without a statistically significant correlation [31].

In this study sample, the respondents considered that the most attractive smile involved a covering of 0.57 mm (in average) of the upper incisors by the upper lip. The differences between the most or least attractive gingival exposure perceived by the participants were not significant according to gender, or between dentists and laypersons. However, participants who had an orthodontic treatment in the past considered that a higher amount of gingival exposure was more attractive, than participants who did not have an orthodontic treatment. When it comes to factors that mostly influence smile attractiveness, laypersons chose significantly more frequent white teeth or aligned teeth as being more important for an attractive smile, while dentists chose significantly more frequent a gingival exposure between 0–3 mm as an important factor for an aesthetic smile.

Due to the fact that dentists and laypersons may have different opinions regarding an ideal result of a dental treatment that has aesthetic repercussions, it is important that patients participate in taking decisions when establishing the treatment plan in order to achieve a harmonious smile characterized by the best relationships between the morphology of the teeth, lips and gums. When discussing the results of a dental treatment that can have consequences on patients’ facial appearance, clinicians can use the results of this study in their clinical practice for a better understanding of their patients’ wishes regarding the exposure of gingival smile during smile.

This study had its limitations. First of all, the image used for the assessment of an ideal amount of gingival exposure was of a female patient, and we did not use a separate image of a male patient. The reason for this choice was based on the study of Geron and Atalia (2005) who stated that the female smile is more critically examined [26]. Another limitation is represented by the inability to control the honesty of the respondents. We had to rely on our respondents’ sincerity when completing the questionnaires. However, we believe that this study is a good starting point in studying the ideal amount of gingival exposure for the Romanian population.

## 5. Conclusions

The most attractive smile involved a coverage of 0.57 mm of the upper incisors by the upper lip, white and aligned teeth versus a gingival exposure being the factors that mostly influence laypersons and dentists when assessing a smile.

## Figures and Tables

**Figure 1 medicina-58-01265-f001:**
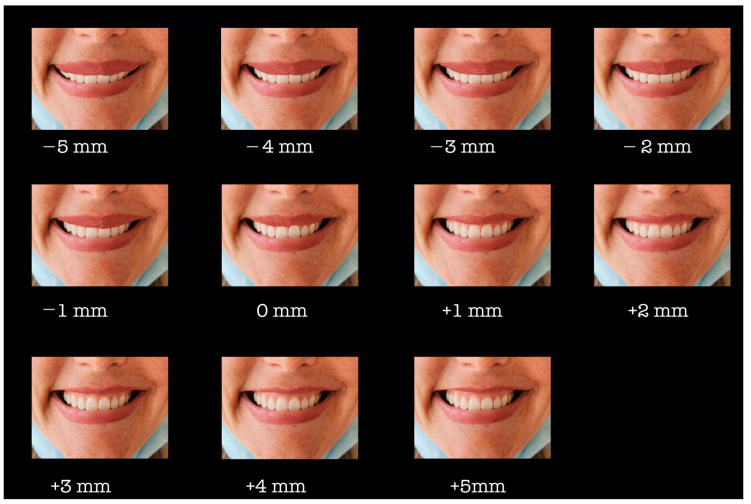
Values of gingival exposure.

**Table 1 medicina-58-01265-t001:** Distribution of the respondents.

Gender.	
Women (No., %)	Men (No., %)
194 (82.6%)	41 (17.4%)
Professional category	
Laypersons (No., %)	Dentists (No., %)
162 (68.9%)	73 (31.1%)
Orthodontic treatment	
Yes (No., %)	No (No., %)
97 (41.3%)	138 (58.7%)

No.—number; %—percentage.

**Table 2 medicina-58-01265-t002:** Average value of gingival exposure for the most/least attractive smile.

Smile Attractiveness	Average Value ± SD (in mm)	Median Value (IQR) (in mm)
Most attractive gingival exposure	−0.57 ± 2.407	0 (−2–1)
Least attractive gingival exposure	1.43 ± 3.785	4 (−2–4)

SD—Standard deviation; IQR—interquartile range.

**Table 3 medicina-58-01265-t003:** Comparison of the most/least attractive gingival exposure between groups.

Group		Most Attractive Gingival Exposure (in mm)	Least Attractive Gingival Exposure (in mm)
		Median (IQR)	Median (IQR)
Women		0 (−2–1)	4 (−2–4)
Men		−1 (−3.5–1)	4 (−3–4)
*p* *		0.090	0.979
Laypersons		0 (−2.25–1)	4 (−2–4)
Dentists		0 (−2–1)	4 (−3.5–4)
*p* *		0.896	0.824
Without orthodontic treatment		0 (−3–1)	4 (0.75–4)
With orthodontic treatment		1 (−2–2)	4 (−5–4)
*p* *		0.009	0.026
Laypersons	Women	0 (−2–1)	4 (−2.25–4)
	Men	−1.5 (−3.75–1)	4 (1–4)
	*p* *	0.232	0.433
Dentists	Women	0 (−2–1)	4 (−1.75–4)
	Men	−1 (−3.5–1)	4 (−5–4)
	*p* *	0.181	0.276
Women	Laypersons	0 (−2–1)	4 (−2.25–4)
	Dentists	0 (−2–1)	4 (−1.75–4)
	*p* *	0.988	0.438
Men	Laypersons	−1.5 (−3.75–1)	4 (1–4)
	Dentists	−1 (−3.5–1)	4 (−5–4)
	*p* *	0.857	0.324

IQR—interquartile range; * Mann-Whitney U Test.

**Table 4 medicina-58-01265-t004:** Distribution of the patients for answers provided for the last item.

Factors	No., %
Aligned teeth	106 (45.1%)
White teeth	50 (21.3%)
Symmetrical smile	41 (17.4%)
0–3 mm gingival exposure	27 (11.5%)
<0 mm gingival exposure	9 (3.8%)
>3 mm gingival exposure	1 (0.4%)
Diastema	1 (0.4%)

No—Number; %—percentage.

**Table 5 medicina-58-01265-t005:** Distribution of the participants between smile factors and other parameters.

Group/Smile Factors		White Teeth	Aligned Teeth	Symmetrical Smile	0–3 mm Gingival Exposure	<0 mm Gingival Exposure	*p* *
Women		39 (78%)	89 (84%)	32 (78%)	24 (88.9%)	8 (88.9%)	0.703
Men		11 (22%)	17 (16%)	9 (22%)	3 (11.1%)	1 (11.1%)	
Laypersons		48 (96%)	76 (71.7%)	24 (58.5%)	8 (29.6%)	5 (55.6%)	<0.001
Dentists		2 (4%)	30 (28.3%)	17 (41.5%)	19 (70.4%)	4 (44.4%)	
Laypersons	Women	37 (77.1%)	66 (86.8%)	18 (75%)	7 (87.5%)	5 (100%)	0.428
	Men	11 (22.9%)	10 (13.2%)	6 (25%)	1 (12.5%)	0 (0%)	
Dentists	Women	2 (100%)	23 (76.7%)	14 (82.4%)	17 (89.5%)	3 (75%)	0.768
	Men	0 (0%)	7 (23.3%)	3 (17.6%)	2 (10.5%)	1 (25%)	
Women	Laypersons	37 (94.9%)	66 (74.2%)	18 (56.3%)	7 (29.2%)	5 (62.5%)	<0.001
	Dentists	2 (5.1%)	23 (25.8%)	14 (43.8%)	17 (70.8%)	3 (37.5%)	
Men	Laypersons	11 (100%)	10 (58.8%)	6 (66.7%)	1 (33.3%)	0 (0%)	0.017
	Dentists	0 (0%)	7 (41.2%)	3 (33.3%)	2 (66.7%)	1 (100%)	
Without orthodontic treatment		33 (66%)	62 (58.5%)	23 (56.1%)	15 (55.6%)	4 (44.4%)	0.715
With orthodontic treatment		17 (34%)	44 (41.5%)	18 (43.9%)	12 (44.4%)	5 (55.6%)	

* Fisher’s Exact Test.

## Data Availability

The data presented in this study are available on request from the corresponding authors. The data are not publicly available due to privacy reasons.

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
