# Peer review of "The Influence of Gingival Exposure on Smile Attractiveness as Perceived by Dentists and Laypersons"

_medicina, 2022, doi:10.3390/medicina58091265_

Round 1

Reviewer 1 Report

Very interest paper. the self perception is individual and the authors discuss about this. there is no pattern about self perception.

Author Response

Dear Reviewer,

This study is of great importance in clinical practice for dentists, especially Romanian dentists, because patients and dentists tend to have different opinions on how a smile should look alike compared to dentists.

Thank a lot for your comments.

            Best wishes,

                        Authors

Reviewer 2 Report

Thank you very much for submitting the above manuscript to the Journal.
While we certainly find the topic of interest, we also find that your study does not provide sufficient novel information to merit publication

We did not find any novel thing that you proposed, compared to previous studies e.g. 

https://www.researchgate.net/publication/323855531_THE_INFLUENCE_OF_GINGIVAL_EXPOSURE_ON_SMILE_ATTRACTIVENESS

http://jurnal.pdgi.or.id/index.php/jida/article/view/670/449

https://www.sciencedirect.com/science/article/abs/pii/S0889540618309314

Could you please highlight more the research gap?

thank you

Author Response

Dear Reviewer,

This study is of great importance in clinical practice for dentists, especially Romanian dentists, because patients and dentists tend to have different opinions on how a smile should look alike compared to dentists. The latter must take into account and put into practice patients` wishes so as the patients are happy with their smile at the end of the treatment. Moreover, this is the first research that highlights the effect of an increased gingival exposure on smile attractiveness from Romanian lay-persons´ and dentists´ point of view.

.           Best wishes,

                        Authors

Reviewer 3 Report

1. In Introduction section please add benefit  of this study in clinical practice 

2. What is the sampling method of this study?--> online survey to which population  please clarify about this

3.How did the the author reduce possible bias in this study? , For examples  the bias due to only specific age group or population who can access the online survey

4. tabel  3 and 5  are not easy to read -->  the p value writing very confusing please make it more easy to read

5. In discussion, it  can be added the comparison  of the result in romanian  population and  with other population. Is there any differences and what is the possible cause of the differences

6. In discussion can be added  about how the clinician can use the result of this study in their clinical practice.

5.The conclusion is to long, It is better make it short and only specify to the important result. The others can be add in the discussion.

Author Response

Dear Reviewer,

This study is of great importance in clinical practice for dentists, especially Romanian dentists, because patients and dentists tend to have different opinions on how a smile should look alike compared to dentists. The latter must take into account and put into practice patients` wishes so as the patients are happy with their smile at the end of the treatment. Moreover, this is the first research that highlights the effect of an increased gingival exposure on smile attractiveness from Romanian lay-persons´ and dentists´ point of view.

This study was addressed to respondents aged over 18 of Romanian ethnicity, who have access to social media, with or without dental studies (if they are dentists, minimum 5 years experience), who might need orthodontic treatment and seem interested in improving their facial appearance.  

We made some changes to table 3 and 5 so as to be easier to read.

When discussing the results of a dental treatment that can have consequences on patients` facial appearance, clinicians can use the results of this study in their clinical practice for a better understanding of their patients` wishes regarding the exposure of gingival smile during smile.

            We reduced the conclusion and exposed only the important results.

            Thanks a lot for your relevant remarks,

Authors

Round 2

Reviewer 2 Report

all the comments addressed well